# Opinion: Establishing a Science-into-Policy Process for Tropospheric Ozone Assessment

Richard G. Derwent[1,*], David D. Parrish[2] and Ian C. Faloona[3]

[1] rdscientific, Newbury, Berkshire, UK

[2] David.D.Parrish, LLC, 4630 MacArthur Ln, Boulder, Colorado, USA

[3] Department of Land, Air, & Water Resources, University of California, Davis, California, USA

*Correspondence to*: Richard G. Derwent (r.derwent@btopenworld.com); David D. Parrish (david.d.parrish.llc@gmail.com)

**Abstract.** Elevated tropospheric ozone concentrations driven by anthropogenic precursor emissions is an environmental hazard scientifically similar to the depletion of the stratospheric ozone layer and global climate change; however, the tropospheric ozone issue lacks the generally accepted, international assessment efforts that have greatly informed our understanding of the other two. Here we briefly review those successful science-into-policy approaches, and outline the elements required to conduct a

similar process for tropospheric ozone. Particular emphasis is placed on the need for establishing a conceptual model to fully understand the underpinning science, useful policy metrics, and motivating international policy forums for regulating anthropogenic ozone production over the hemispheric and global scales, thereby expanding beyond the traditional regional, air basin approach that has dominated air quality regulatory philosophy to date.

## 1 Introduction

The global environmental policy problems involving the depletion of the stratospheric ozone layer and global climate change have been identified, fully researched and moved into their respective policy arenas over the last fifty years or so. A number of features of the ozone depletion and climate change problems that have brought them to the forefront of environmental policy can be detailed and then compared with

25 the corresponding features of the tropospheric ozone problem. The process of science-into-policy is much less developed for tropospheric ozone, as it seems to have fallen between the two stools of ozone depletion and climate change.

While not presenting an existential crisis of the same magnitude as depletion of the stratospheric ozone layer or global climate change, tropospheric ozone is widely recognized as an important air pollutant

30 (Monks et al., 2015). Beyond fundamentally controlling the oxidation potential of Earth's inhabited atmosphere, tropospheric ozone damages human health (Fleming et al., 2018), contributes to the global burden of disease (Cohen et al., 2017), impacts crops and vegetation (Mills et al., 2018; Feng, et al., 2022) and is a man-made greenhouse gas, third in importance behind carbon dioxide and methane (IPCC, 2014; Skeie, et al., 2020).

35 Urban and regional ozone has been subject to policy actions for several decades to reduce air pollution and ozone episodes and these measures have been largely successful in North America (Parrish et al., 2022) and Europe (Derwent and Parrish, 2022), and are making progress in other continents. An important counter example to note here is the growth of tropospheric ozone over East Asia, where in spite of large recent reductions in air pollutant emissions, regional ozone has generally risen (Wang et al., 2020).

40 Regardless of the policy actions, exceedances of air quality standards and guidelines set to protect human health still occur and will do so for the foreseeable future. This is because ozone episodes sit on top of a baseline that is hemispheric and even global in scale. Furthermore, as the manifold, deleterious effects of ozone continue to be revealed by ongoing research, the policy targets of exposure are likely to be reduced even further or different exposure metrics developed. In either case the relative importance of the

45 background is going to become the dominant effect on future compliance or non-compliance.

Here, we outline a process by which understanding of the science underpinning tropospheric ozone could lead to robust international policy action with a view to simultaneously reducing the global scale climate impacts of tropospheric ozone, reaching healthy air quality, and ameliorating damage to crops and vegetation.

## 2 Science-into-policy processes of stratospheric ozone layer depletion

Concerns were first raised in the 1970s about the possible impacts of man-made chlorofluorocarbons (CFCs) on stratospheric ozone by Molina and Rowland (1974). These concerns were put into sharp focus by the discovery of the stratospheric ozone hole in the 1980s in Antarctica by Farman et al. (1985). Policy action followed swiftly, not simply because of the importance of these discoveries, but because a number of potential policy hurdles or stumbling blocks had already been surmounted.

A 'model' had been developed describing the mechanism by which the stratospheric ozone hole formed that was widely accepted by the atmospheric science community (Solomon et al., 1986; Crutzen and Arnold, 1986; Cox and Hayman, 1988). It was proposed that man-made CFCs are photolyzed in the stratosphere to form active chlorine atoms and radicals which catalyze the ozone destruction. Armed with this 'model', the process of review and assessment began in earnest under the aegis of the World Meteorological Organization (WMO). A policy metric was developed, the ozone depletion potential (ODP), for the CFCs and ultimately for all ozone-depleting substances. It should be noted that ODPs cannot be 'observed'. These policy metrics required the 'model' of stratospheric ozone to be well understood so that they could be faithfully derived.

Policy actions were formulated within the Vienna Convention for the Protection of the Ozone Layer under the auspices of the United Nations Environment Program (UNEP). The Vienna Convention used the WMO reviews and assessments to build a set of Protocols identifying each ozone depleting substance in turn and moved towards their phase-out, in order of importance, as determined by the products of the ODPs and the abundances.

## 3 Science-into-policy processes of global climate change

It has been postulated over the last two centuries by Eunice Foote, Joseph Fourier and John Tyndall that carbon dioxide would act as a greenhouse gas (Royal Institution, 2019) and Svante Arrhenius (1896) quantified the global temperature increase that would result from increased $CO_2$ levels. Charles Keeling identified the global scale rise in atmospheric carbon dioxide levels from his observations on Mauna Loa, Hawaii and at the South Pole in late 1950s (Keeling et al., 1989). Policymakers were first made aware of the emerging issue of global climate change in the 1980s. Scientific review and assessment began around

this time under the Intergovernmental Panel on Climate Change (IPCC), spear-headed by Bert Bolin who placed the greatest emphasis on assessment with a clear focus on science-into-policy. The IPCC was formed under the aegis of the WMO and its first scientific assessment was published in 1990 with subsequent major scientific assessment reports on the physical science basis in 1995 (SAR), 2001 (TAR), 2007 (AR4), 2014 (AR5), and 2022 (AR6).

Daniel Albritton (Birks et al., 1992) conceptualized a 'model' describing the basic scientific framework for the IPCC. It describes how atmospheric composition change drives radiative forcing, which in turns drives atmospheric responses in terms of changes in temperature, winds and rainfall (physical responses). These atmospheric responses drive changes in the climate (climate responses) which impact on the biosphere (biological responses) ultimately leading to ecosystem responses, as illustrated in Figure 1a. Feedbacks occur when atmospheric responses, such as melting ice, modify radiative forcing, or when ecosystem responses change atmospheric composition through, for example, changes in wetland methane emissions. In this 'model', global climate change is seen as a system of forcings and feedbacks, with man-made composition change as the ultimate driving force. This system has been represented with increasing sophistication through the development of increasingly complex earth-system models (ESMs).

The series of IPCC reports has been presented to the United Nations Framework Convention on Climate Change (UN FCCC). In response to requests from the UN FCCC, the IPCC developed a policy metric, the global warming potential (GWP), so that the different propensities of a range of different trace gases to influence climate change could be represented on a common basis in policy contexts. Using GWPs the UN FCCC put together a basket of six trace gases and began developing strategies with the aim of reducing dangerous anthropogenic climate change. The basket, however, does not address tropospheric ozone, although it is the third most important man-made greenhouse gas after $CO_2$ and methane (Stevenson et al., 2013). Indeed, tropospheric ozone is one of the most important short-lived climate forcers (SLCF).

**4 Essential elements of the science-into-policy process**

From the two sections above, we can identify the essential elements of the science-into-policy process addressing ozone layer depletion and global climate change, as illustrated in Figure 1b. These are, firstly,

review and assessment of the underpinning science with strict and open peer review and encouragement of research with a clear focus on the science-into-policy process. Secondly, development of a hierarchy of models of the underpinning science, and thirdly, development of a policy metric with full buy-in from the atmospheric science community. But fourthly, and most importantly, we can identify the importance of having an international convention which brings together the policy-making and atmospheric science communities and provides a framework for taking account of both scientific and policy developments.

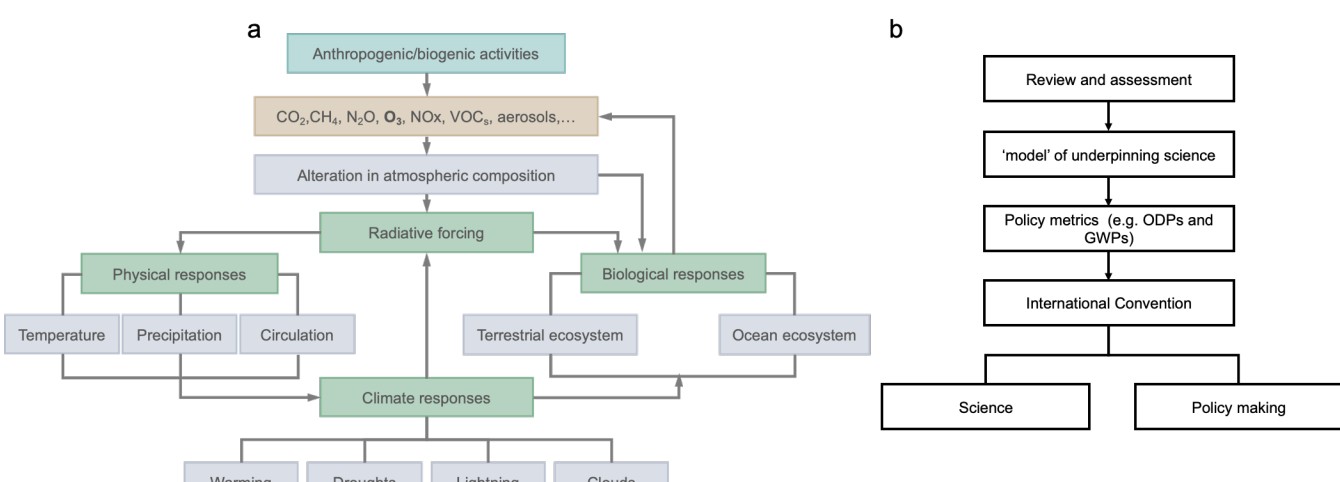

**Figure 1: Science-into-policy processes of Global climate change.** Panel (a) A 'model' of underpinned science representing the key processes involved with climate change revised from Danial Albritton (Birks et al., 1992). Forcings are represented by downwards arrows. Feedbacks are processes by which responses lower down in the diagram drive changes further up. Panel (b) Essential elements of the science-into-policy processes.

## 5 Review and assessment of tropospheric ozone

There are open and accessible data repositories covering many country-wide or regional monitoring networks, for example, those operated by the United States Environmental Protection Agency (US EPA), the United Nations Economic Commission for Europe (UN ECE), the European Monitoring and Evaluation Program (EMEP), etc. In addition, there are some data portals which provide access to network

data, including those operated by the Norwegian Institute for Air Research, the International Global Atmospheric Chemistry (IGAC) Tropospheric Ozone Assessment Report (TOAR) and the WMO World Data Centre for Greenhouse Gases. Some illustrative examples of long-term changes of both baseline and urban ozone concentrations are included in Figure 2. Importantly, the interpretation of baseline ozone and its long-term changes remains an open scientific question.

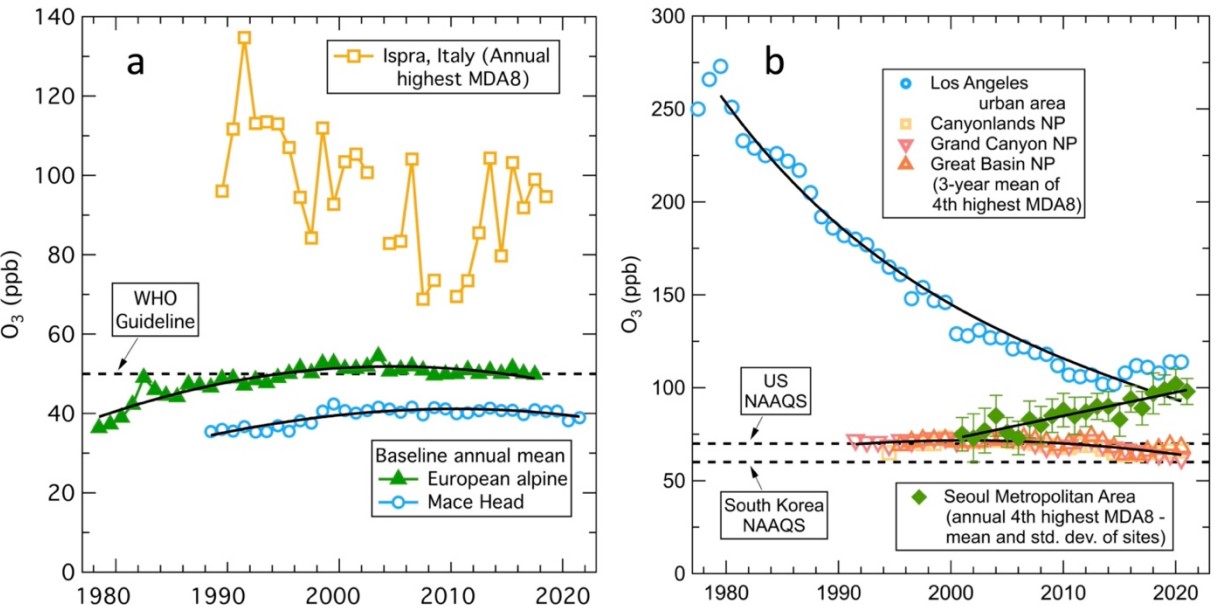

**Figure 2: Example long-term records of measured ozone.** Baseline (Europe and US) and urban (Europe, US and Asia) ozone concentrations are included. Note that differing statistical metrics are used between the records. Data sources: Ispra, Italy and the European alpine sites (the latter described in detail by Parrish, et al., 2020) from the TOAR database (https://join.fz-juelich.de); Mace Head from Derwent et al. (2023); US data sets from the US EPA AQS data archive (https://www.epa.gov/aqs); South Korea from Kim et al. (2022). The trend lines are quadratic polynomial fits to the baseline ozone data sets, a linear fit to the South Korean data, and an exponential decrease above a baseline trend for the Los Angeles data (Parrish et al., 2022).

There are reviews of urban and regional ozone compiled by several organizations including the US EPA, AQEG, EU and EMEP, but they are generally restricted to single jurisdictions or single networks. Hemispheric and global scale reviews are compiled by the UN ECE Task Force on Hemispheric Air

Pollution (HTAP) and TOAR. All these reviews, whilst providing excellent coverage, suffer distinctly from a dichotomous consideration of urban and regional scales on the one hand and the hemispheric and global scales on the other, and lack a strategic focus on science-into-policy for tropospheric ozone mitigation issues.

In response to the human health impacts of elevated ozone levels, policymakers have taken extensive measures on local and regional scales to control the emissions of the main ozone precursors: oxides of nitrogen ($NO_x$) and reactive organic compounds (Sillman et al., 1999; Ehlers et al., 2016; Lu et al., 2010). Control of motor vehicle exhaust emissions through the mandatory fitting of exhaust gas catalysts and evaporative cannisters has largely been a complete success, enabling large reductions to be achieved in ozone precursor emissions from road transport. Elevated ozone levels have declined in almost all legislative and administrative regions where ozone precursor emissions have been controlled. However, these declines have not gone far enough. Air quality targets, guidelines or standards set to protect human health have not always been achieved and, with the identification of adverse health impacts at lower ozone concentrations, there is growing pressure to further tighten standards; with some exceptions, non-attainment of air quality targets remains an important policy issue. Whilst assessment of long-term changes in urban and regional ozone levels point to the huge impact of the measures taken to reduce ozone precursor emissions, they also demonstrate that reductions in exceedances have slowed during the past decade (Parrish et al., 2022; Derwent and Parrish, 2022). The situation is also reflected by the atmospheric chemistry literature which is characterized by dichotomous views separated by scales; with papers on urban ozone (Ehlers et al., 2016; Nelson et al., 2021; Tan et al., 2019; Cardelino and Chameides, 1995; Brune et al., 2016; Pusede et al., 2015) often ignoring or oversimplifying the impact of the background troposphere on the urban environment and papers on the background troposphere (Parrish et al., 2009; Wang and Jacob, 1998; Gaudel et al., 2018) often ignoring or oversimplifying the impact of urban and regional pollution.

## 6 Development of a Hierarchy of 'models' to understand the essential science of tropospheric ozone

A hierarchy of models of the underlying science is required to provide detailed advice for the science-into-policy process. There are excellent reviews and intercomparisons of complex ozone models on all scales from urban to regional to global, addressing issues of atmospheric chemistry, boundary layer processes and atmospheric transport (Simon et al., 2012; Turnock et al., 2020; Young et al., 2013). However, there is also a requirement for conceptual models that aim to advance our understanding of

tropospheric ozone by simplifying and capturing the essence of the most salient physical and chemical processes that control observed ozone abundances. These refined models must be consistent with findings from observations and capture the overall behavior of more complex models. Such simplified presentations are required to facilitate joint communication between different scientific communities and policymakers, and can be instrumental in the continued development of the most complex chemical

transport models.

   Development of a widely-accepted, simple, conceptual 'model' that intuitively explains the broad features of how ozone sources, sinks and transport processes all interact to establish the observed local, regional and large-scale spatial distributions, seasonal cycles and long-term temporal changes of ozone is urgently required. Such a 'model' would form the core of a robust assessment, would be invaluable to

researchers in their efforts to understand the beautifully detailed observational data and chemistry-transport model results that are presently available to the atmospheric community. Such a conceptual 'model' would provide a firm foundation upon which to conceive and organize present and future research efforts into a more comprehensive understanding of all aspects of the spatial and temporal distribution of tropospheric ozone. Such an intuitive model would be an essential component of a modeling hierarchy,

similar to those employed by the geophysical fluid dynamics community (Held, 2005), serving to complement the comprehensive numerical models that aim to simulate in full detail as much of the atmospheric chemistry, dynamics, and coupling thereof as possible.

   Figure 3 presents a schematic diagram illustrating some of the basic principles of the tropospheric ozone issue updated from Derwent et al. (1998b). The diagram envisages background air containing $O_3$, CO and

$CH_4$ entering an urban area or rural region on the right-hand side. Urban and regional biogenic precursor

emissions drive local and regional scale photochemical ozone production which elevates ozone concentrations above the global baseline levels, leading to human health effects and crop and vegetation damage. After one to several days travel downwind, the regionally-polluted air with elevated levels of $O_3$, CO, $NO_x$ and unreacted organic compounds is lofted from the continental boundary layer and rejoins the

global circulation. One aspect of such a 'model' requires particular attention: it should aim to bridge the dichotomous views of urban pollution ignoring or oversimplifying the impact of the background troposphere on the urban environment and the background troposphere ignoring or oversimplifying the impact of urban pollution.

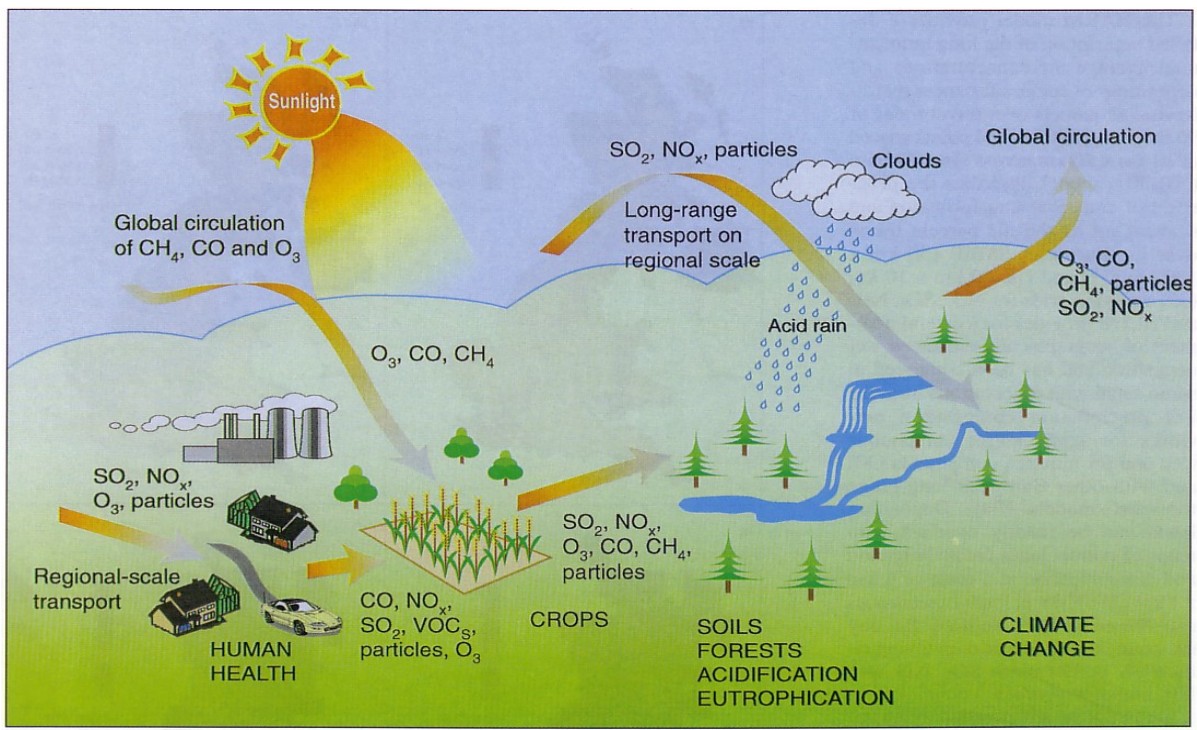

**Figure 3. Schematic illustration of the basic principles underpinning the tropospheric ozone issue.** Note that tropospheric ozone sources include production in background, urban and rural areas as well as injection of ozone from the stratosphere. The ozone produced in the three distinctly different source areas are mixed within the troposphere, which depicts the difficulties of tropospheric ozone control.

Since the manifold of processes determining the tropospheric ozone distribution is considerably simpler

than that driving the climate system, this required 'model' will be simpler than that developed by IPCC.

As examples of its utility, the 'model' should be able to provide a clear description of northern midlatitude baseline ozone, including answering 1) why ozone maximizes in late spring in the free troposphere but peaks earlier in the spring with a summer minimum in the marine boundary layer, 2) why mean concentrations approximately doubled during the first half of the 20$^{th}$ century but have since decreased and 3) to what extent is ozone homogeneously mixed within the prevailing zonal flow. Moreover, the 'model' should explain how ozone compares between the Northern and Southern Hemispheres, both at present and in the pre-industrial atmosphere. Of great utility would be an interactive atlas, such as provided by the IPCC AR6 Synthesis Report, that quantifies background and anthropogenic contributions to ozone concentrations at any specified location on the globe, particularly if those quantities could be illustrated as a function of variable local or hemispheric precursor emissions.

A further important aspect of the 'model' of tropospheric ozone is a close association with the 'model' of climate change. For example, future changes in stratosphere-troposphere exchange due to an accelerated Brewer-Dobson circulation (Abalos et al., 2020) along with a changing Northern Hemisphere stratospheric ozone abundance (Woltmann et al., 2020) may raise background ozone concentrations over the next century. The observed increase in background $NO_x$ (Qu et al., 2021) could be the result of rising soil temperatures, increasing wildfire impacts and increasing lightning (Murray, 2018) production in a warming climate.

## 7 Policy-relevant metrics

With the science-into-policy processes for stratospheric ozone and climate change, the scientific community developed the ODP and GWP metrics. These metrics provided a scientific focus for the actions of policymakers, which necessarily focused attention on the emissions of ozone-depleting substances and greenhouse gases, decreasing the emissions sooner and more strongly for species with the largest metric values.

Developing a policy metric like ODPs and GWPs for tropospheric ozone is more complex. The previously developed parameters such as OFPs (Ozone Formation Potentials) (Carter et al., 1995) and POCPs (Photochemical Ozone Creation Potentials) (Derwent et al., 1998a) are not considered to be quite

comparable with ODPs and GWPs as they are highly state-dependent and so their efficacy as accurate metrics depends on local conditions, which vary over time and space.

A large variety of metrics have been proposed for tropospheric ozone, addressing its impacts on urban and regional air quality and impacts on human health and crops and vegetation (e.g., Monks et al., 2015). Indeed, Lefohn et al. (2018) propose 25 metrics in all, (4 for model-measurement intercomparison, 5 for characterization of ozone in the free troposphere, 11 for human health impacts and 5 for vegetation impacts). The choice of metric for our proposed science-into-policy process for tropospheric ozone will be an important discussion between the scientific and policy communities, and will likely include discussion of complicating issues, as has been the case for global climate change (e.g., Lynch et al., 2020) and stratospheric ozone depletion (e.g., Pyle et al., 2022).

However, as with the stratospheric ozone and climate change issues, policymakers will necessarily focus their attention on controlling the emissions of ozone precursor gases, decreasing the emissions sooner and more strongly for those ozone precursor emissions which more readily affect tropospheric ozone. Ozone precursor emission inventories have been established over at least five decades and have driven policy focus throughout the world. They can provide the essential policy focus for the tropospheric ozone issue.

Regional air quality and global models all require emission inventories (e.g., Figure 4) and chemical mechanisms, together with observations to evaluate their performance. These models effectively convert ozone precursor emission inventory data into predictions of ozone concentrations over specific spatial and temporal scales in response to policy needs. By changing the ozone precursor emissions, whilst keeping all other input data the same, modelers can visualize the impact of particular ozone precursor emission control strategies on the tropospheric ozone distribution, for the benefit of policymakers.

Almost all urban areas and regions have been inventoried in greater or lesser detail for each of the major ozone precursors. Examples include the CHIEF (Clearinghouse for Inventories and Emissions Factors by US EPA), EMEP (by UN ECE), MEIC (Multi-resolution Emission Inventory for China), SAFAR (System of Air Quality and Weather Forecasting and Research), EDGAR (Emission Database for Global Atmospheric Research, from IGAC) and CEDS (Community Emission Data System) (Hoesly et al., 2018).

Whilst the emission factor approach is well-defined for man-made emissions, a different approach is
required for biogenic emissions. Accurate estimation of the biogenic emissions of isoprene, terpenes and
NO$_x$ requires information on plant and tree species, on ecosystem composition and on the local
meteorological conditions of temperature, radiation and soil moisture. Biomass burning is another source
difficult to treat rigorously in ozone policy models; it is important to treat both agricultural biomass
burning and wild fires separately for policy purposes.

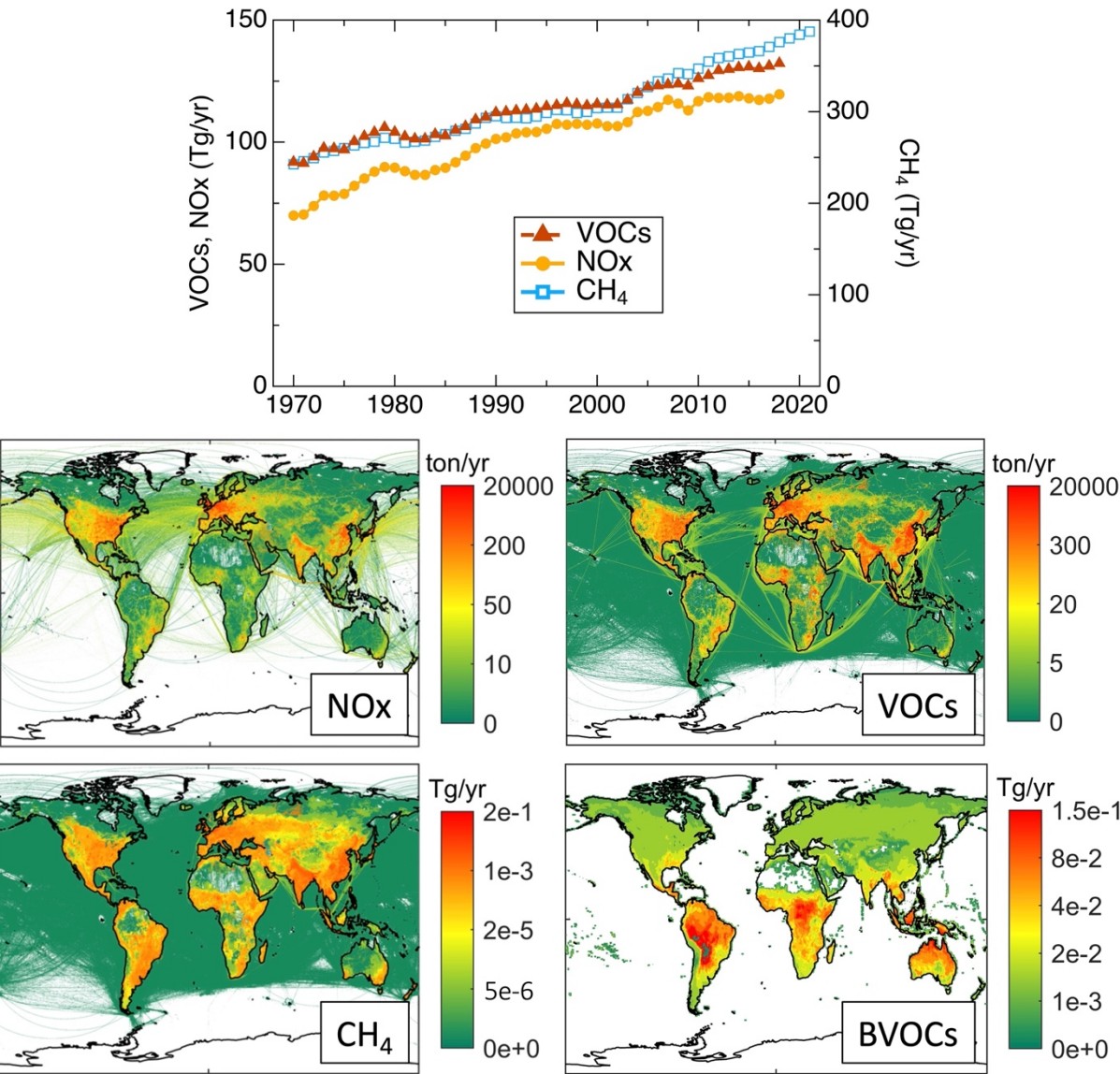

**Figure 4. Emission rates of key ozone precursors.** (Top) Time series of annual emission rates of anthropogenic $NO_x$ and VOCs on the left axis and total $CH_4$ on the right axis. (Middle) Spatial distributions of mean annual emissions of $NO_x$ and VOCs (metric ton/yr/0.1 deg x 0.1 deg) over 1984–2015. (bottom) Spatial distributions of mean annual emissions of $CH_4$ and BVOCs (Tg/yr/0.1 deg x 0.1 deg) over 1984–2020. Underlying emission data are from EDGARv6.1, EDGARv7.0 and MEGAN-MACC. Total annual mean BVOCs emissions are ~200 Tg/yr, with no significant trend.

In summary, we consider that ozone precursor emission inventories can provide an up-to-the-task basis for driving policy formulation for tropospheric ozone. High annual emission rates (> 100 tonne/yr/grid square) over densely populated regions (e.g., Asia, Europe and North America) dominate the total emission of global anthropogenic $NO_x$ and VOCs (Figure 4) which supports the conclusions that intensive anthropogenic air pollutant emissions in these regions make great contributions to the non-attainment of ozone air quality standards. Moreover, $CH_4$ emissions increased similarly to those of $NO_x$ and VOCs (Figure 4) also indicating possible accelerated photochemical $O_3$ production in background air masses. Importantly, these inventories require continual updating and extension to additional species (e.g., volatile chemical products, McDonald et al., 2018), emission sectors and global regions; "top-down" evaluation is essential as emission sources evolve (e.g., McDonald et al., 2018; Smith et al., 2022).

## 8 Implications for future policy and science

Whilst there are many international bodies that address issues relevant to improving scientific knowledge of the sources and distribution of tropospheric ozone, there is no international convention that could readily take up this issue and make policy progress globally. HTAP considers regions covering only Europe, USA and Canada. It specifically excludes Asia, Mexico and North Africa, together with Southern Hemisphere countries.

The IPCC has identified tropospheric ozone as the third most important man-made greenhouse gas after carbon dioxide and methane. The latter trace gas, methane, has been established as an important tropospheric ozone precursor (e.g., West et al., 2013). The 6th IPCC Assessment Report in its Working Group I report addressed tropospheric ozone as a Short-lived Climate Forcer and stressed the co-benefits

of methane reductions to mitigate climate change and improve air quality. The Working Group II report underlines the effects of ozone on crops and warns of the health dangers of elevated ozone levels during heat waves. The Working Group III report dealt with decarbonizing strategies for tackling climate change and the co-benefits for ozone air quality.

The UN FCCC has pioneered the compilation of ozone precursor emission inventories from each country globally but does not include tropospheric ozone in its basket of trace gases and there has been little focus on tropospheric ozone within the UN FCCC. If SLCFs could be moved up the UN FCCC agenda and policy actions focused on methane and tropospheric ozone, a rapid impact on climate forcing would result because of the relatively short global mean atmospheric lifetimes of both methane (12 years) and tropospheric ozone (~1 month). There would also be a substantial improvement in urban and regional ozone air quality, bringing the possibility of achieving air quality standards and guidelines set to protect human health and crops and vegetation.

There are some crucial scientific issues that need resolution ahead of the proposed science-into-policy process and the development of policy advice leading to the regulation of tropospheric ozone. There is a conflict in the assessment of current ozone observations between two viewpoints concerning baseline ozone trends in the northern midlatitudes; namely, is baseline ozone continuing to rise (Gaudel et al., 2018; Tarasick et al., 2019) or has baseline ozone peaked during the 2000s and 2010s and is beginning to decline (Logan et al., 2012; Parrish et al. 2021). Because urban and regional ozone episodes sit on top of a hemispheric-scale ozone background, any trend in background ozone has a direct influence on the attainment of policy goals based on the achievement of air quality standards or guidelines set for the protection of human health.

The majority of these crucial scientific issues involve ozone modelling from the urban through to the global scale and, in particular, their uncertainties as quantified in international model inter-comparison exercises. The magnitude of model uncertainties imply that models should be regarded as indicative rather than prescriptive policy tools. As with climate science and climate models (Carslaw et al., 2018), there are areas where model uncertainties are apparent and where further scientific study aimed at reducing uncertainties should be encouraged. For tropospheric ozone, these include:

- Ozone trends since pre-industrial times (since these fix the radiative forcing from ozone; Stevenson et al., 2013), trends since the 1950s (covering the period of instrumental ozone observations; Parrish et al., 2014) and since the 1990s (covering the period of intense ozone monitoring).

- Ozone seasonal cycles and interhemispheric gradients (Derwent et al., 2016).

- Intercontinental ozone precursor source-receptor relationships for methane, carbon monoxide, $NO_x$ and VOCs linking to receptors in regional monitoring stations across the northern hemisphere continents (Fiore et al., 2009; HTAP, 2010).

- Biomass burning and wild fires as sources of tropospheric ozone (Jaffe et al., 2020) and air quality standard and guideline exceedance.

- The impact of climate change on the strength of the Brewer-Dobson circulation in the stratosphere and the consequences for the stratosphere-troposphere exchange as a source of tropospheric ozone (Abalos et al., 2020).

We have every confidence that if a robust and cogent, peer-reviewed policy-oriented scientific review of the tropospheric ozone issue could be assembled by the atmospheric science community, then policy

progress could be made under the auspices of the UNEP. Such a review could be built on the current TOAR and HTAP activities, with inclusion of participation by policy associated scientists. Coordinated global action on tropospheric ozone holds the promise of delivering acceptable ozone air quality in all major population and industrial centers globally, a prospect that is unlikely to be achievable without such collective planning. Furthermore, future actions to reduce urban and regional ozone precursor emissions

may be wasteful of resources if realistic account of the hemispheric or global baseline is not taken. In such a situation the exceedance of air quality standards and guidelines will continue unchecked and the hands of local policy makers will be tied.

**Acknowledgements**

Discussions with Keding Lu, Alex Archibald, David Stevenson, Daniel Jacob, Tao Wang, Ken Carslaw,

Paul Monks, Jim Crawford and Martin Schultz are kindly acknowledged; they contributed to the

discussions that led to the writing of the article, although they do not necessarily agree with the content. I.C.F.'s effort was supported by the USDA National Institute of Food and Agriculture (Hatch project CA-D-LAW-2481-H).

## Competing Interests

The contact authors have declared that none of the authors has any competing interests.

## Author contributions

R.G.D. and D.D.P. were responsible for the overall design. R.G.D. and D.D.P. wrote the paper with input from I.C.F.

## Data availability

Annual highest MDA8 ozone data at Ispra, Italy and the annual mean ozone data at European alpine sites (described in detail by Parrish, et al., 2020) are available from the TOAR Surface Ozone Database via the JOIN web interface: https://join.fz-juelich.de. Baseline annual mean ozone data from Mace Head are taken from Derwent et al. (2023). The 3-year mean of 4th highest MDA8 (i.e., the US ozone design value) from the four US data sets in Figure 2 are available from the US EPA AQS data archive

(https://www.epa.gov/aqs). The surface monitor data for South Korea are available from https://www.airkorea.or.kr/web/; the annual 4th highest MDA8 data for the Seoul Metropolitan Area are taken from Kim et al. (2022). Globally-averaged annual mean and $0.1° \times 0.1°$ grid maps of $NO_x$ and VOCs data are obtained from EDGARv6.1 and EDGARv7.0 Global Air Pollutant Emissions databases (https://edgar.jrc.ec.europa.eu/index.php/dataset_ap61; https://edgar.jrc.ec.europa.eu/dataset_ghg70).

Global annual BVOC data are taken from MEGAN-MACC Biogenic emission inventory (Sindelarova et al., 2014), which are available in ECCAD database (https://eccad.aeris-data.fr/).

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
