# Peer review of "Opinion: Establishing a Science-into-Policy Process for Tropospheric Ozone Assessment"

_EGUsphere, 2023_

## Referee Comment (RC2)

**Review of "Opinion: Establishing a Science-into-Policy Process for Tropospheric Ozone Assessment" by Derwent et al.**

In this Opinion piece, Derwent et al. seek to highlight that the existential problem of increases in the abundance of tropospheric ozone could be addressed through a more formal science-to-policy framework – the likes of which has had success in protecting the stratospheric ozone layer and (hopefully) limiting the effects of anthropogenic climate change.

Clearly everyone is entitled to their own opinions and the authors are world renowned and respected for their work on tropospheric ozone, so their opinions matter. But I feel there are some significant issues with this piece that should be addressed before final publication. The major issues are discussed below alongside more minor issues latter.

Major issues:
   1) **Do we not already have a "model" of tropospheric ozone?**
A key premiss of this paper is that as a community we lack a parsimonious model that can describe the processes that control tropospheric ozone. Although the level of simplicity is arguable, at least in my mind we have such a parsimonious model. Indeed, Figure 3 in the paper outlines such a model and this model has been the *de facto* model used within the community since at least the mid 2000s. In which case, what new insight is this Opinion piece adding?

At its simplest we can say that the model of climate change is a question of forcing and feedback:

$\Delta N = \Delta F - \alpha \Delta T$                                              **Equation 1**

The model of tropospheric ozone can also be written very simply (where $\nabla$ is used to represent transport, $P(O_3)$ the production and $L$ the first-order loss rate of ozone):

$P(O_3) = (L + \nabla)[O_3]$                                              **Equation 2**

However, these simple models are not practically useful. Complex problems require complex models. There is a good point to be made that the level of complexity of our model (Figure 3) is not fit for purpose but it's not clear how we as a community go about determining this. It seems to me, at least, that the model we have for tropospheric ozone (Figure 3) is fine. The main problem is the problem of who owns the challenge of tropospheric ozone (the air quality community or the climate community) and so who are we simplifying the model (Figure 3) for; this is an issue that is intimately linked with the choice of metric.

   2) **Do we have a process for deciding which metrics for tropospheric ozone are policy relevant?**
Section 7, I think, is a key section for this Opinion piece. The authors outline some of the metrics used in the climate science and stratospheric ozone communities (GWP and ODP) and some of those used in the tropospheric chemistry community (OFP and POCP) but the authors don't go on to highlight the problems with the GWP and ODP metrics. A discussion on the problems with these metrics would be helpful as that would help underscore the

need for a process to develop the optimal policy relevant metric(s) for tropospheric ozone. See for example, Lynch et al. (2020) and Pyle et al. (2022).

The discussion about the UN FCCC is important (not necessarily interesting) but the UN FCCC deals with emitted species only, as these emissions can be regulated. Should the UN FCCC also consider OH as one of the gases it "controls"? Tropospheric ozone cannot be part of emission based policy metrics because it is not an emitted species. The UN FCCC does include methane and a significant fraction of the methane GWP comes from the impacts that methane has on tropospheric ozone. If tropospheric ozone were to come under the remit of UN FCCC then the fraction of GWP that is attributable to tropospheric ozone formation from methane would have to be removed. This would create a huge issue in terms of recent work that targets methane mitigation as a priority as the GWP-100 of methane would drop by about ¼. Again, a discussion of the impacts of the choice of policy metric would really help the community rally around a process to identify the right one(s).

Figure 2 highlights the alarming issue we have with metrics for tropospheric ozone. By my counting there are at least 4 different metrics being displayed. I think that an Opinion piece such as this should touch on this important aspect and draw on the literature which has discussed the choice of metrics at length. Through analysis of this literature it rapidly becomes evident that part of the problem with creating a "simple" model for tropospheric ozone is that the stakeholders for the impacts of tropospheric ozone are diverse and each want different things. A key and related aspect is which policy makers are the metrics being targeted at? Policy is a wide ranging world and many different tropospheric ozone metrics could be identified for different policy issues. This relates to my point about who owns the challenge of tropospheric ozone above.

**Minor points:**
L94: I suggest you delete the word "Interestingly" and let the reader make up their mind.
L115: The heading seems incomplete or at least it does to me. Delete "the" or add more words.
L129: I'm sure there are others but with my UK-centric hat on I would suggest you add AQEG to this list who have done fantastic work on tropospheric ozone for decades.
L183: See major comments above.
Figure 4: Methane emissions should top out at about 500 Tg/yr. Please check panel (a). The use of NMVOC and AVOC is confusing. Can you be consistent and define what you mean here. Also, please check the units for panels (b)-(e). Should there not be an area dimension?
L240: Fragment. Re-word.
L242: Replace the comma with a semi-colon or re-phrase the sentence here.
L255: Add "e.g.," to the reference as this was not the first study to point this out.
L260&266: What do the authors mean by "ozone air quality" and "air quality for ozone"?

**References:**

Lynch, J., Cain, M., Pierrehumbert, R. and Allen, M., 2020. Demonstrating GWP*: a means of reporting warming-equivalent emissions that captures the contrasting impacts of short-and long-lived climate pollutants. Environmental Research Letters, 15(4), p.044023.

Pyle, J.A., Keeble, J., Abraham, N.L., Chipperfield, M.P. and Griffiths, P.T., 2022. Integrated ozone depletion as a metric for ozone recovery. Nature, 608(7924), pp.719-723.

---

## Community Comment (CC4)

**TOAR (Tropospheric Ozone Assessment Report) Steering Committee (past and present) comment on:**

**Opinion: Establishing a Science-into-Policy Process for Tropospheric Ozone Assessment**
Richard G. Derwent, David D. Parrish and Ian C. Faloona
submitted to ACPD

The following is a joint statement from current and former members of the Tropospheric Ozone Assessment Report (TOAR) Steering Committee.  The submitted manuscript proposes a science-into-policy process that would mis-interpret the findings from TOAR, and therefore we feel compelled to state our concerns regarding the scientific structure of the proposal.

- The submitted manuscript makes no mention of IPCC's well-known assessment of the co-benefits of greenhouse gas mitigation for air quality improvements, a concept that has been widely discussed by the atmospheric sciences community and by policy-makers for at least 10 years (e.g. see West et al., 2013; 391 citations according to Web of Science).  As summarized by the recent Synthesis Report of IPCC AR6 (https://report.ipcc.ch/ar6syr/pdf/IPCC_AR6_SYR_LongerReport.pdf), existing and new policies to reduce greenhouse gas emissions will have the co-benefit of reducing ozone at the surface and in the free troposphere, especially due to methane mitigation (see also IPCC AR6 WG-III).  This omission of IPCC findings is profound, and seriously undermines the suggested science to policy process. The authors also fail to discuss the inclusion of tropospheric ozone as a risk factor in recent Global Burden of Disease reports, which have brought tropospheric ozone into the public health community discourse (Murray et al. Lancet. 2020; 396: 1223-1249).

- The submitted manuscript calls for the development of a single ozone policy metric, "with full buy-in from the atmospheric science community".  TOAR is a grassroots organization sustained by the atmospheric science community, and TOAR's great success is due to its popular and necessary use of multiple ozone metrics (for climate, health and vegetation impacts).  Tropospheric ozone chemistry is extremely complicated, concentrations of ozone vary in space and on hourly timescales, and no single ozone metric can adequately gauge its impacts on diverse biological systems, or climate.  The suggestion for a single ozone policy metric would not provide protection for the different receptors damaged by ozone which have very different exposure patterns, dose-response curves, and ozone damage thresholds.  Rather than a new metric which would necessitate the development of a new set of exposure-response curves, ozone policy could be guided by the more consistent use of existing response curves to convert exposure or dose to easily understood impacts such as years of life lost (YLLs), years of life lived with disability (YLDs), and disability-adjusted life-years (DALYs), crop production losses (CPL) and economic cost losses (ECL).

- A foundation of this proposal is the authors' repeated claim that mid-latitude baseline ozone doubled from the 1950s to the early 2000s, but has since been steadily decreasing.  This claim runs contrary to the findings of IPCC AR6 and other recent assessments of tropospheric ozone trends, including the analyses from TOAR (collectively cited over 1300 times), which do not support a steady decrease in tropospheric ozone across the mid-latitudes in recent decades (further details are provided below).  This basic scientific error prevents us from having any confidence in the scientific structure of the proposed science-into-policy process.

- These authors call for the development of a simple, conceptual 'model' that would be used to understand the output of atmospheric chemistry models, guide research efforts and inform policy. They describe the attributes of the "model", which exactly match the attributes of a conceptual model that

these same authors have proposed in a recent paper (Mims et al. 2022). While the authors do not cite their own work, we briefly discuss the weaknesses of the Mims et al. model below.  In our expert opinion, output from modern atmospheric chemistry models can be effectively summarized for policy-makers, and there is no reasonable application for a simple, conceptual model that lacks basic atmospheric dynamics and is therefore unable to capture the temporal and spatial variability in column and ground level ozone, let alone allow for any attribution of ozone changes to driving forces.  There may be important roles for simple models, but new models must be vetted among the community of scientists and demonstrate their value before they are used in a science-to-policy process.

While we agree that science must inform policy, we have no confidence in this particular proposal for a science-into-policy process, which seems to oversimplify the science and relevant metrics, while misinterpreting the science.  TOAR follows the lead of other influential scientific processes like IPCC (which focuses on the science and summarizing that science for policymakers), to inform choices without prescribing policy.  TOAR does so in part by including studies of impacts on health, crops, vegetation, and climate.  TOAR will continue to work with IPCC, the Climate and Clean Air Coalition (CCAC, www.ccacoalition.org) and the Task Force on Hemispheric Transport of Air Pollution (TF HTAP) under the UNECE, as well as established regional organizations (for example, EMEP in Europe), to advise policy-makers to develop more effective approaches.

Signed:
**TOAR-II co-chairs:**
**Dr. Martin G. Schultz**, Jülich Supercomputing Centre, Forschungszentrum Jülich, Germany, TOAR co-chair 2014-present
**Dr. Helen Worden**, Atmospheric Chemistry Observations and Modeling Laboratory (ACOM), National Center for Atmospheric Research (NCAR), Boulder, CO, USA, TOAR-II co-chair 2022-present

**Members of the TOAR steering committee (past and present):**
**Dr. Owen R. Cooper**, CIRES University of Colorado Boulder/NOAA CSL, former TOAR co-Chair 2014-2022
**Prof. Lisa Emberson**, Environment & Geography Dept., University of York, York, U.K.
**Prof. Mat Evans**, Wolfson Atmospheric Chemistry Laboratories, Dept Chemistry, University of York, York, UK
**Prof. Zhaozhong Feng**, School of Applied Meteorology, Nanjing University of Information Science & Technology, Nanjing 210044, China
**Dr. Jacek W. Kaminski**, WxPrime Corporation, Toronto, Canada
**Dr. Yugo Kanaya**, Earth Surface System Research Center (ESS), Japan Agency for Marine-Earth Science and Technology (JAMSTEC), Yokohama, Kanagawa 2360001, Japan
**Dr Raeesa Moolla**, School of Geography, Archaeology and Environmental Studies, University of the Witwatersrand, Johannesburg, South Africa
**Dr. Manish Naja**,  Aryabhatta Research Institute of Observational Sciences (ARIES), Manora Peak, Nainital - 263 001, INDIA
**Dr. Elena Paoletti** IRET CNR via Madonna del Piano 10, 50019 Sesto Fiorentino, Italy
**Dr. Gabriele Pfister**, Senior Scientist, Atmospheric Chemistry Observations and Modeling Lab (ACOM), National Center for Atmospheric Research (NCAR), Boulder CO
**Yinon Rudich**, Department of Earth and Planetary Sciences, Weizmann Institute, Rehovot 76100, Israel
**Dr. Rodrigo J. Seguel**, Center for Climate and Resilience Research/Department of Geophysics, University of Chile
**Dr. Baerbel Sinha**, Department of Earth and Environmental Sciences, Indian Institute of Science Education and Research Mohali, India

**Dr. David W. Tarasick**, Environment and Climate Change Canada, 4905 Dufferin Street, Downsview, ON, M3H 5T4 Canada

**Dr. Anne M. Thompson**, Senior Scientist Emeritus, NASA/Goddard Space Flight Center, Greenbelt, MD 20771 USA; Senior Research Faculty, University of Maryland-Baltimore County, Baltimore MD 21228

**Dr. Erika von Schneidemesser**, Research Institute for Sustainability, Potsdam, Germany

**Prof. J. Jason West**, University of North Carolina at Chapel Hill

**Prof. Lin Zhang**, Department of Atmospheric and Oceanic Sciences, School of Physics, Peking University, Beijing 100871, China
* * *
**Supporting information:**

**Simple conceptual 'model'**

As stated above, the submitted manuscript calls for the development of a simple, conceptual 'model' that would be used to understand the output of atmospheric chemistry models, guide research efforts and inform policy. They describe the attributes of the "model", which exactly match the attributes of a conceptual model that these same authors have proposed in a recent paper (Mims et al. 2022). Even though the authors do not cite their own work, we briefly comment on this paper in order to point out the substantial shortcomings of a simple conceptual model. This particular conceptual model is similar to a simple 1970s box model that scientists had to build in the days before adequate computing power was available to run more complex models (e.g. Oeschger et al., 1975; Thompson and Schneider, 1979). It has no atmospheric dynamics and it assumes the mid-latitudes are isolated from the polar regions and the tropics; this is contrary to recent work, which shows that tropospheric ozone in the mid-latitudes is impacted by emissions and transport from the tropics, and this influence cannot be ignored (Zhang et al., 2016,2021; Gaudel et al., 2020). In contrast, modern atmospheric chemistry models can handle global and regional atmospheric dynamics, in addition to emissions and photochemistry. These models correctly reproduce the observed increase of the tropospheric ozone burden, and as shown by IPCC AR6 the output from these models can be effectively summarized to provide the answers to the questions from policy makers (see Chapters 6 and 7, and Box TS.7 in the Technical summary of AR6).

**Baseline ozone trends:**

As assessed by IPCC AR6 WG-I (Chapters 2 and 6), the annual State of the Climate Reports, the Tropospheric Ozone Assessment Report (Tarasick and Galbally et al., 2019), CMIP6 and the UNEP Scientific Assessment of Ozone Depletion 2022 (Chapter 3.3) the tropospheric ozone burden has continued to increase since the 1990s including at mid-latitudes; these same assessments found no convincing evidence that mid-latitude baseline ozone doubled from the 1950s to the early 2000s. These findings are corroborated by very recent studies published since the release of IPCC AR6 (Miyazaki et al., 2020; Christensen et al., 2022; Wang et al., 2022; Fiore et al., 2022; Chang et al., 2022). Contrary to the evidence, the authors of the submitted manuscript have claimed (since at least 2017) that baseline ozone has been consistently decreasing across northern mid-latitudes over the past two decades. No independent study has been able to corroborate their claims, and these claims were not accepted by the assessment reports listed above.

**References**

Chang, K.-L., et al. (2022), Impact of the COVID-19 economic downturn on tropospheric ozone trends: an uncertainty weighted data synthesis for quantifying regional anomalies above western North America and Europe, *AGU Advances, 3*, e2021AV000542. https://doi.org/10.1029/2021AV000542

Christiansen, A., Mickley, L. J., Liu, J., Oman, L. D., and Hu, L.: Multidecadal increases in global tropospheric ozone derived from ozonesonde and surface site observations: can models reproduce ozone trends?, Atmos. Chem. Phys., 22, 14751–14782, https://doi.org/10.5194/acp-22-14751-2022, 2022.

Fiore, Arlene M., et al. (2022), Understanding recent tropospheric ozone trends in the context of large internal variability: A new perspective from chemistry-climate model ensembles, *Environmental Research: Climate, https://doi.org/10.1088/2752-5295/ac9cc2*

Gaudel, A., et al. (2018), Tropospheric Ozone Assessment Report: Present-day distribution and trends of tropospheric ozone relevant to climate and global atmospheric chemistry model evaluation, Elem. Sci. Anth., 6(1):39, DOI: https://doi.org/10.1525/elementa.291

Gaudel, A., O. R. Cooper, K.-L. Chang, I. Bourgeois, J. R. Ziemke, S. A. Strode, L. D. Oman, P. Sellitto, P. Nédélec, R. Blot, V. Thouret, C. Granier (2020), Aircraft observations since the 1990s reveal increases of tropospheric ozone at multiple locations across the Northern Hemisphere. Sci. Adv. 6, eaba8272, DOI: 10.1126/sciadv.aba8272

Mims, C.A., **Parrish, D.D., Derwent, R.G**., Astaneh, M. and **Faloona, I.C**., 2022. A conceptual model of northern midlatitude tropospheric ozone. *Environmental Science: Atmospheres*, *2*(6), pp.1303-1313.

Miyazaki, K., Bowman, K., Sekiya, T., Eskes, H., Boersma, F., Worden, H., Livesey, N., Payne, V.H., Sudo, K., Kanaya, Y. and Takigawa, M., 2020. Updated tropospheric chemistry reanalysis and emission estimates, TCR-2, for 2005–2018. *Earth System Science Data*, *12*(3), pp.2223-2259.

Oeschger, H., Siegenthaler, U., Schotterer, U. and Gugelmann, A., 1975. A box diffusion model to study the carbon dioxide exchange in nature. *Tellus*, *27*(2), pp.168-192.

Tarasick, D. W., I. E. Galbally, O. R. Cooper, M. G. Schultz, G. Ancellet, T. Leblanc, T. J. Wallington, J. Ziemke, X. Liu, M. Steinbacher, J. Staehelin, C. Vigouroux, J. W. Hannigan, O. García, G. Foret, P. Zanis, E. Weatherhead, I. Petropavlovskikh, H. Worden, M. Osman, J. Liu, K.-L. Chang, A. Gaudel, M. Lin, M. Granados-Muñoz, A. M. Thompson, S. J. Oltmans, J. Cuesta, G. Dufour, V. Thouret, B. Hassler, T. Trickl and J. L. Neu (2019), Tropospheric Ozone Assessment Report: Tropospheric ozone from 1877 to 2016, observed levels, trends and uncertainties. Elem Sci Anth, 7(1), DOI: http://doi.org/10.1525/elementa.376

Thompson, S.L. and Schneider, S.H., 1979. A seasonal zonal energy balance climate model with an interactive lower layer. *Journal of Geophysical Research: Oceans*, *84*(C5), pp.2401-2414.

Wang, H., Lu, X., Jacob, D. J., Cooper, O. R., Chang, K.-L., Li, K., Gao, M., Liu, Y., Sheng, B., Wu, K., Wu, T., Zhang, J., Sauvage, B., Nédélec, P., Blot, R., and Fan, S. (2022), Global tropospheric ozone trends, attributions, and radiative impacts in 1995–2017: an integrated analysis using aircraft (IAGOS) observations, ozonesonde, and multi-decadal chemical model simulations, Atmos. Chem. Phys., 22, 13753–13782, https://doi.org/10.5194/acp-22-13753-2022

West, J.J., Smith, S.J., Silva, R.A., Naik, V., Zhang, Y., Adelman, Z., Fry, M.M., Anenberg, S., Horowitz, L.W. and Lamarque, J.F., 2013. Co-benefits of mitigating global greenhouse gas emissions for future air quality and human health. *Nature climate change*, *3*(10), pp.885-889.

Zhang, Y., O. R. Cooper, A. Gaudel, A. M. Thompson, P. Nédélec, S.-Y. Ogino and J. J. West (2016), Tropospheric ozone change from 1980 to 2010 dominated by equatorward redistribution of emissions, Nature Geoscience, 9(12), p.875, doi: 10.1038/NGEO2827

Zhang, Y., West, J.J., Emmons, L.K., Flemming, J., Jonson, J.E., Lund, M.T., Sekiya, T., Sudo, K., Gaudel, A., Chang, K.L. and Nédélec, P., 2021. Contributions of world regions to the global tropospheric

ozone burden change from 1980 to 2010. Geophysical Research Letters, 48(1), p.e2020GL089184.

---

## Author Response (AR1)

**RESPONSE TO REVIEWs OF MANUSCRIPT-egusphere-2023-426 - Opinion: Establishing a Science-into-Policy Process for Tropospheric Ozone Assessment**

**GENERAL RESPONSES TO THE REVIEWS:**

**POINT-BY-POINT RESPONSES TO REVIEWER COMMENTS:**

**Overview of our point-by-point responses:**

In the following, the referees' individual comments are reproduced, and each is followed by text in blue italics giving our response. Each also describes any additions/revisions to the manuscript relevant to that comment.

**CC1: 'Comment on egusphere-2023-426 - Martin Schultz'**

Having been acknowledged for "helpful discussions" in this article, I would like to make it very clear that I don't agree with the opinions that are put forward by the authors. This approach has several flaws. In particular, the idea that "simple models" can advance our understanding and help shaping a better environmental policy, is complete nonsense.

*We thank Martin Schultz for initiating the discussion of our recently posted Opinion. In acknowledging helpful discussions, we did not mean to imply that all were supportive; indeed, constructively critical comments are often the most helpful. Notwithstanding his curt opinion regarding "simple models" being "complete nonsense", we would like to point out that scientists in other fields have expressed much more supportive views of the important roles played by models of varying complexity. In fact, in the field of geophysical fluid mechanics it is quite common for researchers to rely on a variety of numerical models of differing complexity and representation to investigate the manifold, non-linear features of atmospheric motion.*

*No changes have been made to the text.*

**RC1: 'Comment on egusphere-2023-426', Anonymous Referee #1**

This manuscript addresses a need of a science-into-policy approach for tropospheric ozone management. Tropospheric ozone is an essential oxidant modulating tropospheric chemistry and is an effective greenhouse gas. Near surface ozone is an air pollutant that is harmful to public health and vegetation. There are currently active discussions of lowering air quality standard for ozone, which causes concerns about increased non-attainments because of large background ozone values. Therefore, it is timely to publish an opinion article that suggests actions to better understand and reduce tropospheric ozone. The authors reviewed the stratospheric ozone layer depletion and global climate change topics as examples of the science-into-policy process and suggest a similar process for tropospheric ozone.

I think highly of this opinion that outlines the processes from the review and assessment to an international convention and that explains the need of this approach. The authors pointed out several important issues of tropospheric ozone for international research such as local anthropogenic emission changes, background ozone trends associated with global emissions and climate changes (biogenic emissions, lightning, wildfires), and stratosphere-troposphere exchanges. I would recommend to publish this article and promote discussions about tropospheric ozone and methane as the UN FCCC agenda and policy actions.

Development of a "model" of the underpinning science for tropospheric ozone would be challenging. According to the manuscript, the "model" needs to be widely-accepted, simple, conceptual and intuitively explains the broad features of tropospheric ozone including chemical sources, sinks, and transport processes and local, regional, and large-scale spatial and temporal distributions (including long-term trends). And this model plays an important role in a robust assessment. To my opinion, such a conceptual model would not be highly accurate. But, the model (or model development process) is still helpful to identify essential factors determining tropospheric ozone distributions, to calculate ozone budgets and to initiate discussions advancing tropospheric ozone science and policy at the same ground. This model can be regarded as one simple tool or reference.

*Thank you for your supportive comments on the role of a simplified conceptual model. A reference has been added in Section 4 to a hierarchy of models and a paragraph has been added in Section 6 to explain that the simplified model lies alongside the complex models and is not to be considered a replacement of the complex models currently in use.*

This opinion would be an excellent starting point to discuss about more organized and supported international efforts to diagnose tropospheric ozone problems and develop "science-to-policy" processes to reduce tropospheric ozone.

Minor change

P3, L74: Correct "International Panel on Climate Change" to "Intergovernmental Panel on Climate Change".

*We thank Referee #1 for carefully reading our Opinion and for their positive comments and recommendation (https://doi.org/10.5194/egusphere-2023-426-RC1).*

*"Development of a "model" of the underpinning science for tropospheric ozone" likely will indeed be challenging, as evidenced by the comment posted by the TOAR Steering Committee (https://doi.org/10.5194/egusphere-2023-426-CC4). In our response to that comment (https://doi.org/10.5194/egusphere-2023-426-AC5) we more fully describe the "model" that we envision; it will necessarily comprise a hierarchy of models of varying complexity. This issue is now more fully described in our revised manuscript.*

*The suggested correction of "International Panel on Climate Change" to "Intergovernmental Panel on Climate Change" has been made in our revised manuscript.*

**RC2: 'Comment on egusphere-2023-426', Anonymous Referee #2**

In this Opinion piece, Derwent et al. seek to highlight that the existential problem of increases in the abundance of tropospheric ozone could be addressed through a more formal science-to-policy framework – the likes of which has had success in protecting the stratospheric ozone layer and (hopefully) limiting the effects of anthropogenic climate change.

Clearly everyone is entitled to their own opinions and the authors are world renowned and respected for their work on tropospheric ozone, so their opinions matter. But I feel there are some significant issues with this piece that should be addressed before final publication. The major issues are discussed below alongside more minor issues later.

*Thank you for your supportive comments.*

Major issues: 1) Do we not already have a "model" of tropospheric ozone?

A key premiss of this paper is that as a community we lack a parsimonious model that can describe the processes that control tropospheric ozone. Although the level of simplicity is arguable, at least in my mind we have such a parsimonious model. Indeed, Figure 3 in the paper outlines such a model and this model has been the de facto model used within the community since at least the mid-2000s. In which case, what new insight is this Opinion piece adding?

At its simplest we can say that the model of climate change is a question of forcing and feedback:

$\Delta N = \Delta F - a\Delta T$          Equation 1

The model of tropospheric ozone can also be written very simply (where $\nabla$ is used to represent transport, P(O3) the production and L the first-order loss rate of ozone):

$P(O3) = (L + \nabla)[O3]$      Equation 2

However, these simple models are not practically useful. Complex problems require complex models. There is a good point to be made that the level of complexity of our model (Figure 3) is not fit for purpose but it's not clear how we as a community go about determining this. It seems to me, at least, that the model we have for tropospheric ozone (Figure 3) is fine.

*Thank you for these comments about simple, conceptual models and for your thoughts about two simple models for climate and ozone based on equations 1 and 2. They do clearly illustrate the limitations of simplified, conceptual models. Your proposed simple equation is perhaps a first step in determining that the level of our proposed "simplified" model would need to include more than merely photochemical production/loss and transport of autogenous ozone. In fact, there would need to be two critical "forcing" terms: stratospheric inputs and surface deposition, which can be time dependent and variable across the globe. The former is independent of the tropospheric ozone burden, the latter proportional to surface ozone concentration but also strongly dependent on how the terrestrial biosphere changes in our changing climate. We therefore conclude that, no, the proposed model would need to be more complex than Equation 2 above, but this is the beginning of the conversation we were hoping to start with this piece. We feel that simple (meaning simpler than full-blown chemical-transport models), conceptual models have an important role in stimulating discussions between the scientific and policy communities on tropospheric*

*ozone. A reference has been added in Section 4 to the required hierarchy of models and a paragraph has been added in Section 6 using largely your above comments to explain that the simplified model lies alongside the complex models and is not a replacement.*

The main problem is the problem of who owns the challenge of tropospheric ozone (the air quality community or the climate community) and so who are we simplifying the model (Figure 3) for; this is an issue that is intimately linked with the choice of metric.

2) Do we have a process for deciding which metrics for tropospheric ozone are policy relevant?

Section 7, I think, is a key section for this Opinion piece. The authors outline some of the metrics used in the climate science and stratospheric ozone communities (GWP and ODP) and some of those used in the tropospheric chemistry community (OFP and POCP) but the authors don't go on to highlight the problems with the GWP and ODP metrics. A discussion on the problems with these metrics would be helpful as that would help underscore the need for a process to develop the optimal policy relevant metric(s) for tropospheric ozone. See for example, Lynch et al. (2020) and Pyle et al. (2022).

*Thank you for these thoughtful comments on policy metrics. The climate change and ozone-depletion issues have simple policy metrics, and we are looking for a similar metric for the proposed science-into-policy process for tropospheric ozone. Policymakers are working satisfactorily with ozone precursor emission inventories to develop and promulgate policy proposals and initiatives for urban and regional ozone. We think that this would be a satisfactory approach for tropospheric ozone, generally. The text has been amended to include a brief discussion of possible coming difficulties and have included the Lynch et al. (2020) and Pyle et al. (2022) references.*

The discussion about the UN FCCC is important (not necessarily interesting) but the UN FCCC deals with emitted species only, as these emissions can be regulated. Should the UN FCCC also consider OH as one of the gases it "controls"? Tropospheric ozone cannot be part of emission-based policy metrics because it is not an emitted species. The UN FCCC does include methane and a significant fraction of the methane GWP comes from the impacts that methane has on tropospheric ozone. If tropospheric ozone were to come under the remit of UN FCCC then the fraction of GWP that is attributable to tropospheric ozone formation from methane would have to be removed. This would create a huge issue in terms of recent work that targets methane mitigation as a priority as the GWP-100 of methane would drop by about ¼. Again, a discussion of the impacts of the choice of policy metric would really help the community rally around a process to identify the right one(s).

*The UN FCCC, as we explain in the text, develops its policies using GWPs as metrics and uses greenhouse gas emissions as the instrument of policy control. Although tropospheric ozone is predominantly a secondary pollutant, we are proposing that ozone precursor emissions could be the instrument of global policy control as it is now, but on a country-by-country basis.*

*We do not see why the development of the proposed ozone policy would necessitate the removal of the methane impacts on ozone in the calculation of its GWP. Just as most air pollution policy excludes methane as an ozone precursor, the same could hold for the UN FCCC policy, albeit with the understanding that the budgets of methane and ozone are*

*photochemically interconnected. If tropospheric ozone came under the UN FCCC, there would be no question of changing the GWP for methane. The GWP is a theoretical quantity, based on atmospheric science and its magnitude would not change if tropospheric ozone came under the UN FCCC.*

*No changes were made to the text.*

Figure 2 highlights the alarming issue we have with metrics for tropospheric ozone. By my counting there are at least 4 different metrics being displayed. I think that an Opinion piece such as this should touch on this important aspect and draw on the literature which has discussed the choice of metrics at length. Through analysis of this literature it rapidly becomes evident that part of the problem with creating a "simple" model for tropospheric ozone is that the stakeholders for the impacts of tropospheric ozone are diverse and each want different things. A key and related aspect is which policy makers are the metrics being targeted at? Policy is a wide-ranging world and many different tropospheric ozone metrics could be identified for different policy issues. This relates to my point about who owns the challenge of tropospheric ozone above.

*Thank you for these thoughtful comments on policy metrics. The climate change and ozone-depletion issues have simple policy metrics, and we are looking for a similar metric for the proposed science-into-policy process for tropospheric ozone. As we note above, policymakers are working satisfactorily with ozone precursor emission inventories to develop and promulgate policy proposals and initiatives for urban and regional ozone. We think that this would be a satisfactory approach for tropospheric ozone, generally. The text has not been amended.*

Minor points:

L94: I suggest you delete the word "Interestingly" and let the reader make up their mind. L115: The heading seems incomplete or at least it does to me. Delete "the" or add more words.

L129: I'm sure there are others but with my UK-centric hat on I would suggest you add AQEG to this list who have done fantastic work on tropospheric ozone for decades.

L183: See major comments above.

Figure 4: Methane emissions should top out at about 500 Tg/yr. Please check panel (a). The use of NMVOC and AVOC is confusing. Can you be consistent and define what you mean here. Also, please check the units for panels (b)-(e). Should there not be an area dimension?

L240: Fragment. Re-word.

L242: Replace the comma with a semi-colon or re-phrase the sentence here.

L255: Add "e.g.," to the reference as this was not the first study to point this out.

L260&266: What do the authors mean by "ozone air quality" and "air quality for ozone"?

*Thank you for these comments and corrections. In all cases the text has been amended accordingly.*

**CC2**: 'Comment on egusphere-2023-426', Sophie Szopa

**CC3**: 'Reply on CC2', Sophie Szopa

Climate change and air pollution are both critical environmental issues that are already affecting humanity. In its 6th assessment cycle, the Intergovernmental panel on climate change (IPCC) dedicated a full chapter (chapter 6) in the Working Group I report to Short Lived Climate Forcers, including tropospheric ozone. The evolution of ozone abundance is also assessed in the chapter 2 of the WGI report. The WGI summary for policymakers stresses the co-benefits of methane emission reduction to mitigate climate change and reduce surface ozone and mentions the need to have coordinated climate and air pollution policies. The WGII report also mentions ozone. For example, the chapter dealing with crops underlines the effect of ozone on crop and food security and the chapter on health warns about the possible compounds effect of ozone peaks and heat waves occurring simultaneously. The WGIII report assessed the co-benefit of decarbonization through air pollution reduction (including ozone) and associated economic benefits. Finally, the synthesis report includes explanation on the co-benefit on health and crop due to air pollution reduction and mentions ozone (see section 4.2 of the synthesis report released in march 2023) and recalls that international environment agreement such as those targeting transboundary air pollution may help to stimulate low GHG investment and reduce GHG emissions. The summary for policymakers of the synthesis report mentions the rapid co-benefit on air pollution (and thus on health) obtained with strong reduction of GHG with a particular emphasis on methane but also remind that dedicated air pollution policies can bring results more rapidly. These summaries for policymakers are approved line by line with government delegates and tailored to ensure that robust and relevant science-based information are provided to policy makers. The underlying material is grounded in assessments based on the analysis of thousands of publications with release of several drafts of the reports that can be reviewed by the scientific community to ensure robustness and transparency.

In addition, the difficulty of having climate metrics relevant for SLCFs is also mentioned in WGI chapter 7. The WGI chapter 6 relies on studies using a wide range of models with varying complexities depending on the aspects assessed (see also BOX 6.1 in chapter 6). This diversity of tools is necessary considering the complexity and non-linearity of atmospheric chemistry (see also the BOX 6.2 in chapter 6).

*We thank Sophie Szopa for clearly and concisely summarizing the discussion of tropospheric ozone that has been included in the 6th assessment cycle by the Intergovernmental Panel on Climate Change (IPCC). It will be critically important that the Science-into-Policy process for tropospheric ozone that we propose fully coordinate on the overlapping issues that IPCC has already assessed. We have added a brief discussion to this effect to our revised manuscript in Section 8.*

**CC4**: 'Comment on egusphere-2023-426', Helen Worden

**TOAR (Tropospheric Ozone Assessment Report) Steering Committee (past and present)**

The following is a joint statement from current and former members of the Tropospheric Ozone Assessment Report (TOAR) Steering Committee. The submitted manuscript proposes a science-into-policy process that would mis-interpret the findings from TOAR, and therefore we feel compelled to state our concerns regarding the scientific structure of the proposal.

- The submitted manuscript makes no mention of IPCC's well-known assessment of the co-benefits of greenhouse gas mitigation for air quality improvements, a concept that has been widely discussed by the atmospheric sciences community and by policy-makers for at least 10 years (e.g. see West et al., 2013; 391 citations according to Web of Science). As summarized by the recent Synthesis Report of IPCC AR6 (https://report.ipcc.ch/ar6syr/pdf/IPCC_AR6_SYR_LongerReport.pdf), existing and new policies to reduce greenhouse gas emissions will have the co-benefit of reducing ozone at the surface and in the free troposphere, especially due to methane mitigation (see also IPCC AR6 WG-III). This omission of IPCC findings is profound, and seriously undermines the suggested science to policy process.

*Thank you for this comment. Text has been added to Section 8 to reinforce the contribution played by the IPCC and its assessments.*

The authors also fail to discuss the inclusion of tropospheric ozone as a risk factor in recent Global Burden of Disease reports, which have brought tropospheric ozone into the public health community discourse (Murray et al. Lancet. 2020; 396: 1223-1249).

*We do mention the health effects of tropospheric ozone and point to the Global Burden of Disease through our reference to Cohen et al. (2017), but we do agree that this issue deserves more emphasis, and the text has been modified. The Murray et al. Lancet paper is merely a Viewpoint article which does not give any detailed reference to ozone, so we have not added it to the text.*

- The submitted manuscript calls for the development of a single ozone policy metric, "with full buy-in from the atmospheric science community". TOAR is a grassroots organization sustained by the atmospheric science community, and TOAR's great success is due to its popular and necessary use of multiple ozone metrics (for climate, health and vegetation impacts). Tropospheric ozone chemistry is extremely complicated, concentrations of ozone vary in space and on hourly timescales, and no single ozone metric can adequately gauge its impacts on diverse biological systems, or climate. The suggestion for a single ozone policy metric would not provide protection for the different receptors damaged by ozone which have very different exposure patterns, dose-response curves, and ozone damage thresholds. Rather than a new metric which would necessitate the development of a new set of exposure-response curves, ozone policy could be guided by the more consistent use of existing response curves to convert exposure or dose to easily understood impacts such as years of life lost (YLLs), years of life lived with disability (YLDs), and disability-adjusted life-years (DALYs), crop production losses (CPL) and economic cost losses (ECL).

*Thank you for these thoughtful comments on policy metrics. The climate change and ozone-depletion issues have simple policy metrics, and we are looking for a similar metric for the*

*proposed science-into-policy process for tropospheric ozone. Policymakers are working satisfactorily with ozone precursor emission inventories to develop and promulgate policy proposals and initiatives for urban and regional ozone. We think that this would be a satisfactory approach for tropospheric ozone, generally. Presumably there are second-order models, undergoing continuous refinement, that can map the first-order tropospheric ozone results into any number of exposure metrics (YLLs, YLDs, CPL, and ECL that you mention) for any given particular policy implication. The text has not been amended.*

- A foundation of this proposal is the authors' repeated claim that mid-latitude baseline ozone doubled from the 1950s to the early 2000s, but has since been steadily decreasing. This claim runs contrary to the findings of IPCC AR6 and other recent assessments of tropospheric ozone trends, including the analyses from TOAR (collectively cited over 1300 times), which do not support a steady decrease in tropospheric ozone across the mid-latitudes in recent decades (further details are provided below). This basic scientific error prevents us from having any confidence in the scientific structure of the proposed science-into-policy process.

*We agree that there is disagreement within the community on important aspects of the temporal and spatial distribution of tropospheric ozone, including long-term changes in baseline ozone. However, the analyses showing "mid-latitude baseline ozone doubled from the 1950s to the early 2000s, but has since been steadily decreasing" remains firmly established for the Northern Hemisphere, not having been refuted by any later analyses; a detailed discussion of this issue is given in the Supporting Information of our response to the Comment of Worden (https://doi.org/10.5194/egusphere-2023-426-AC5). Furthermore, we suspect that the TOAR findings cited above are heavily influenced by "confirmation bias"; agreement with model simulations has been a criterion when choosing between observational analyses that give conflicting findings. Moreover, TOAR has the stated aim to provide a reliable historical record of background ozone levels and reliable guidance to other assessments such as IPCC AR6. Community reliance on TOAR analyses is based on this stated aim without further evaluation of the historical record, so that reliance (in the form of >1,300 citations) does not support the accuracy of the TOAR analyses. One effort of the assessment that we propose is a rigorous, objective evaluation of observational analyses free of such biases. In any event, we do not believe that citation count can be used as logical argumentation in the debate of an open scientific question.*

*However, the Opinion does not actually refer to this issue. Nevertheless, it is important and we accept the reviewers' comments. A paragraph on this issue has therefore been added in Section 8, together with four references.*

- These authors call for the development of a simple, conceptual 'model' that would be used to understand the output of atmospheric chemistry models, guide research efforts and inform policy. They describe the attributes of the "model", which exactly match the attributes of a conceptual model that these same authors have proposed in a recent paper (Mims et al. 2022). While the authors do not cite their own work, we briefly discuss the weaknesses of the Mims et al. model below. In our expert opinion, output from modern atmospheric chemistry models can be effectively summarized for policy-makers, and there is no reasonable application for a simple, conceptual model that lacks basic atmospheric

dynamics and is therefore unable to capture the temporal and spatial variability in column and ground level ozone, let alone allow for any attribution of ozone changes to driving forces. There may be important roles for simple models, but new models must be vetted among the community of scientists and demonstrate their value before they are used in a science-to-policy process.

*Thank you for these comments about simple, conceptual models. They further clarify the limitations of such models. Nevertheless, we feel that simple, conceptual models have an important role in stimulating discussions between the scientific and policy communities on tropospheric ozone. A reference has been added in Section 4 to a hierarchy of models and a paragraph has been added in Section 6 using largely your above comments to explain that the simplified model lies alongside the complex models and is not to be considered a replacement.*

While we agree that science must inform policy, we have no confidence in this particular proposal for a science-into-policy process, which seems to oversimplify the science and relevant metrics, while misinterpreting the science. TOAR follows the lead of other influential scientific processes like IPCC (which focuses on the science and summarizing that science for policymakers), to inform choices without prescribing policy. TOAR does so in part by including studies of impacts on health, crops, vegetation, and climate. TOAR will continue to work with IPCC, the Climate and Clean Air Coalition (CCAC, www.ccacoalition.org) and the Task Force on Hemispheric Transport of Air Pollution (TF HTAP) under the UNECE, as well as established regional organizations (for example, EMEP in Europe), to advise policy-makers to develop more effective approaches.

*We regret that the TOAR community has no confidence in our proposed science-into-policy-process despite their huge contribution to furthering understanding of tropospheric ozone. No amendments have been made to the text in response to these comments.*

**POINT-BY-POINT RESPONSES TO EDITOR'S COMMENTS:**

I found the discussion of the paper particularly useful and it changed my presumed understanding of what you were probably trying to explain. I believe that I and some reviewers/commenters initially misunderstood statements of the original paper. The text in lines 156 - 164 gave me the provocative impression that existing complex CTM models are not useful and should be replaced by a single simple model, although I only partially understood the reasoning behind this proposal. After reading the discussion, I believe I understand the following: the simplified model should be consistent with findings from observational data and CTM modeling. It should represent fundamental relationships between tropospheric ozone and influencing key processes reasonably correct, help to understand the key drivers/ processes, illustrate major uncertainties, and help to define research/policy priorities. The simplified presentation should facilitate joint communication between different scientific communities and policy makers. Nevertheless, complex models would still be needed if the full complexity of spatial distributions and temporal variations or small-scale (e.g., city) are to be resolved. If my understanding is correct, you should emphasize more clearly that the proposed model is not intended to be a replacement for current activities, but a complementary approach (interpretative tool?). Perhaps this can be

illustrated with examples? Perhaps, you can also comment on the required level of chemical complexity in a simplified model (e.g., of VOCs in a global, regional, local atmosphere).

*Thank you for these supportive comments. They have helped us to further clarify the role of simplified, conceptual models in the science-into-policy process. A reference has been added in Section 4 to a hierarchy of models and a paragraph has been added in Section 6 using largely your above comments to explain that the simplified model lies alongside the complex models and is not a replacement.*

---

## Author Response (AR2)

**Response To Editor's Comments on Manuscript-Egusphere-2023-426 - Opinion: Establishing a Science-into-Policy Process for Tropospheric Ozone Assessment**

In the following, the editor's individual comments are reproduced, and each is followed by text in *blue italics* giving our response. Each also describes any additions/revisions to the manuscript relevant to that comment.

Please make it clear in the Acknowledgement that the colleagues listed there do not necessarily agree with the content of the article, but have only contributed to the discussions that led to the writing of the article.
*The relevant sentence in the Acknowledgement has been changed to read:*
*"Discussions with Keding Lu, Alex Archibald, David Stevenson, Daniel Jacob, Tao Wang, Ken Carslaw, Paul Monks, Jim Crawford and Martin Schultz are kindly acknowledged; they contributed to the discussions that led to the writing of the article, although they do not necessarily agree with the content."*

There is a controversial debate about the shape of the northern mid-latitude baseline ozone over the last half century that is mentioned in the TOAR's community comment and which you have now explicitly adressed in the revised manuscript on page 14 (lines 301 – 305). However, the temporal trend you advocate (increase in the 20th century until the early 2000s and a decrease thereafter) is introduced and shown earlier in the paper (page 6, Figure 2; page 9, lines 203 - 205), yet without indication that the ozone baseline is interpreted differently by other researchers. The open scientific question of the ozone baseline should already be mentioned at Figure 2.
*The following sentence has been added immediately preceding Figure 2:*
*"Importantly, the interpretation of baseline ozone and its long-term changes remains an open scientific question."*

Furthermore, references should be provided for the data points and trend lines shown in the figure.
*The following sentences have been added to the caption of Figure 2 (note that this duplicates some of the Data availability section):*
*"Data sources: Ispra, Italy and the European alpine sites (the latter described in detail by Parrish, et al., 2020) from the TOAR database (https://join.fz-juelich.de); Mace Head from Derwent et al. (2023); US data sets from the US EPA AQS data archive (https://www.epa.gov/aqs); South Korea from Kim et al. (2022). The trend lines are quadratic polynomial fits to the baseline ozone data sets, a linear fit to the South Korean data, and an exponential decrease above a baseline trend for the Los Angeles data (Parrish et al., 2022).*